# Computational investigation of missense somatic mutations in cancer and potential links to pH-dependence and proteostasis

**Shalaw Sallah, Jim Warwicker** *

Division of Molecular and Cellular Function, Faculty of Biology, Medicine and Health, Manchester Institute of Biotechnology, University of Manchester, Manchester, United Kingdom

* jim.warwicker@manchester.ac.uk

## Abstract

Metabolic changes during tumour development lead to acidification of the extracellular environment and a smaller increase of intracellular pH. Searches for somatic missense mutations that could reveal adaptation to altered pH have focussed on arginine to histidine changes, part of a general arginine depletion that originates from DNA mutational mechanisms. Analysis of mutations to histidine, potentially a simple route to the introduction of pH-sensing, shows no clear biophysical separation overall of subsets that are more and less frequently mutated in cancer genomes. Within the more frequently mutated subset, individual sites predicted to mediate pH-dependence upon mutation include NDST1 (a Golgi-resident heparan sulphate modifying enzyme), the HLA-C chain of MHCI complex, and the water channel AQP-7. Arginine depletion is a general feature that persists in the more frequently mutated subset, and is complemented by over-representation of mutations to lysine. Arginine to lysine balance is a known factor in determining protein solubility, with higher lysine content being more favourable. Proteins with greater change in arginine to lysine balance are enriched for cell periphery location, where proteostasis is likely to be challenged in tumour cells. Somatic missense mutations in a cancer genome number only in the 10s typically, although can be much higher. Whether the altered arginine to lysine balance is of sufficient scale to play a role in tumour development is unknown.

## 1. Introduction

The tumour microenvironment is affected by metabolic changes in cancer cells. Both hypoxia and acidosis have been characterised, with hypoxia a therapeutic target [1, 2]. An increasing interest in acidosis associated with cancer metabolism has the perspective that pH changes are not only the result of altered metabolism, but also regulate tumour progression, and therefore offer therapeutic opportunities [1]. In response to a shift towards glycolytic metabolism and other acid producing processes, acid extruding transporters are up-regulated at multiple levels of expression control, with additional modulation from the pH-sensitivity of glycolytic enzymes and transporters [3]. A role for somatic mutations altering net acid extrusion from

**Data Availability Statement:** All relevant data are within the manuscript.

**Funding:** This work was supported by UK Biotechnology and Biological Sciences Research

Council grant BB/V0065921/1 to JW. The funders had no role in study design, data collection and analysis, decision to publish, or preparation of the manuscript.

**Competing interests:** The authors have declared that no competing interests exist.

cells and oncogenic processes has been discussed in the context of acid-resistant cells being associated with aggressive phenotypes [4]. A survey of tumour somatic mutations in acid-base transporters argued that interpretation would be improved with additional data, for example transporter flux measurement and environmental pH reporters [5]. Mechanisms of pH-sensing in cancer have been placed into the context of a wider role in cellular physiology and pathophysiology [6].

Cancer cells tend to have a higher intracellular pH (pHi) than normal cells (pH ~ 7.2), with a reversed pH gradient across the cancer cell membrane, and an extracellular pH (pHe) less than normal (pH ~ 7.4) [1]. A review of measured pH values shows decreases of 0.3–0.7 in pHe, dependent upon tissue, with the minimum pHe of 6.4 for lung tissue [7]. Changes are smaller for pHi, typically only 0.1 [7]. It is appreciated that within these average values, there are localised regions of acidification in the tumour microenvironment [8].

The relevance of ionisation at the histidine sidechain, and its role in protein structure and function, to pH changes in cancer has been recognised [9]. Databases such as the Catalogue of Somatic Mutations in Cancer (COSMIC) collect and annotate somatic mutations from cancer genome sequencing projects [10]. Using these data, enrichments for mutation from arginine and (to a lesser degree) for mutation from glutamic acid to lysine, have been seen [11–13]. This observation has been termed arginine depletion in human cancers, and related to C > T transitions in nucleotide mutational signatures, with subsequent selection for function at the amino acid (AA) level suggested for some sites [14]. Mutational signatures in cancers have been widely studied since the advent of gene and subsequently genome sequencing methods, and in general terms result from the superposition of specific cancer mutational mechanisms with weaker and constant endogenous mutation processes [15]. An example of a specific cancer mutational signature is the prevalence of C > T and CC > TT mutations at dipyrimidines in skin cancer, which correlates with the effect of UV light [16]. Bioinformatics analysis of cancer genomes has revealed > 100 mutational signatures, which are also being coupled to genome topographical properties in the ENCODE database [17] (for example nucleosome occupancy of base-pairs) [18]. All cancer types include some C > T mutations (at CpG dinucleotides), arising from the normal endogenous process of 5-methylcytosine deamination [15, 16], contributing to Arg depletion at the amino acid level [14].

Proposed functional significance of arginine mutation includes a potential role for cysteine in neutralising reactive oxygen species [13, 19], and pH-sensing for the introduction of histidine [20], with variation of AA mutation frequencies between cancer types noted [11, 14]. Arginine to histidine mutations that mediate altered pH-dependence of activity, and could be related to fitness advantage in cancer cells include EGFR-R776H and p53-R273H [20], and IDH1-R132H where production of the oncometabolite D-2-hydroxyglutarate is rendered pH-dependent if mutant and wild type IDH1 form a heterodimer [21]. Mutation away from histidine can also modulate pH-dependent function, as seen with β-catenin-H36R [22].

Computational analysis of missense somatic mutation, from the COSMIC database, is facilitated by the availability of AlphaFold2 models for protomers [23], for which coverage of the entire proteome comes at the expense of excluding homo and heter-oligomeric structures, other than in specific cases. Methods for prediction of pKas and protein pH-dependence such as PROPKA [24] and pkcalc [25] are sufficiently fast to allow computation for large datasets [26]. Constant pH molecular dynamics techniques explicitly link conformational change to protonation events [27], but are not currently of sufficient speed for application to the human proteome. Both PROPKA3 [28] and pkcalc have been benchmarked against experimental data [25, 29].

This study looks at missense somatic mutations in COSMIC, recapitulating the prominence of mutation from arginine, followed by a focus on solvent accessible surface area (SASA)

calculation and pKa predictions. Mutations involving histidine are the focus, given the proximity of its normal sidechain pKa to pHi and pHe, noting that other AAs can be involved in pH-sensing [30]. Mutations with higher occurrence in COSMIC ($\geq 10$) were assumed to be more likely driver mutations, and they show no systematic deviation of predicted pKa from presumed passenger (1 occurrence) mutations. Specific sites that are of potential interest for pH-sensing are described. Arginine depletion, and a smaller lysine supplementation effect, are discussed in the context of protein solubility, for which arginine/lysine balance is known to contribute. Consideration is given to the numbers of mutations in tumour cells, and the location of the most mutated proteins.

## 2. Materials and methods

Somatic mutations in cancer were obtained from version 97 of the COSMIC database [10], and filtered for missense mutations, to analyse single AA changes. Ensembl transcript identifiers in the COSMIC data were mapped to UniProt [31] identifiers. Redundancy due to multiple entries for the same mutation in a unique combination of COSMIC tumour and sample identifiers was removed. The resulting list was ranked by the number of COSMIC instances (occurrences) for each mutation, where these instances are comparable to the numbers displayed in the Gene View page of the COSMIC database web site. Conversion of identifiers allowed direct matching with AlphaFold2 models for protomers in the human proteome [23], and used the Retrieve/ID mapping facility at UniProt. Subsets of mutations were made according to ranges of instances recorded in COSMIC. The total number of mutations recorded is 3,289,443, the majority of which (2,625,973) have a single instance. The subset of mutations with $\geq 10$ instances (instGE10), used in this study, numbers 9,919. When reduced to being unique by wild type AA, the number of sites is 2,769,929, being 24.3% of the AlphaFold2 structurally annotated human proteome.

Mutation matrices were made for combinations of wild type and target AAs, and presented as heat maps. For sets of COSMIC missense mutations, over all cancer types, the heat map is calculated as percentages of the (20x19) combinations within each set. Distinction by cancer type was introduced using Human Protein Atlas [32] classifications, cross referenced to those in the COSMIC database. The Mutation Assessor (MA) tool was used to retrieve a set of Functional Impact Score data that reflects amino acid conservation [33]. Enrichment of protein subsets in Gene Ontology (GO) terms engaged the Princeton GO Finder tool [34]. Subcellular location of proteins was added from UniProt. Three tools that have been constructed to assess benign versus deleterious mutation effects on protein function were used to analyse a subset of mutations to histidine: PolyPhen-2 [35], SIFT [36], and AlphaMissense [37].

In order to estimate the difference between observed instances of mutation from wild type AAs in COSMIC, and that expected without adaptation, a simple model was constructed. Wild type AA sites with a single mutation in COSMIC were assumed to not be drivers, and thus to represent a background probability for mutation, when combined with the total number of each AA type in proteins that have mutations in COSMIC. This background probability was then propagated successively through increasing mutation instances, differencing between the current number of instances and the next, to give a predicted number of mutations (assuming no growth advantage) for each instance number and wild type AA. Instances were gathered into ranges to examine the results as the instances increase. These data were processed as both absolute numbers, and as the percentage distributions over AA types, within each range of instances. Finally, the predicted background distributions were differenced with the observed distributions.

Structure based calculations were made with locally installed code for SASA (sacalc), and for pKa (pkcalc, PROPKA) [24, 29]. For some cases, molecular graphics representation of predicted pKas are shown using the protein-sol server [25]. Models for single site mutation in an AlphaFold2 protomer were made with SwissModel [38]. In order to make a distinction between relatively buried and accessible AAs, a threshold of 20 Å$^2$ was used.

## 3. Results and discussion

### 3.1 Mutation from arginine is a dominant feature in somatic mutation matrices

A set of 9,919 mutations from the February 2023 (version 97) COSMIC database, with $\geq 10$ instances (instGE10) was selected to represent a set with sufficient numbers for analysis of the 20 x 19 mutation matrix, but also enriched for driver mutations. The most obvious features in the distribution of instGE10 mutations, other than being sparse due to the limited codon changes associated with single nucleotide mutation, are the preponderance of mutations from Arg (to Cys, Gln, His, Trp in particular), from Ala to Thr and Val, and also from Glu to Lys (Fig 1A). When compared (by differencing) with the set of all mutations that occur just once in the February 2023 COSMIC database (presumed passenger, instEQ1), most prominent are from Arg to Cys/Gln mutations (Fig 1B). Mutations from Arg to His for instGE10 are enriched relative to instEQ1, as are Glu to Lys, but not as much as from Arg to Cys/Gln mutations. As reported by others [11–13] mutation from Arg is a major feature of in the landscape of somatic mutations in cancer, but it is also apparent that rather than a specific emphasis on mutation to His in a set that is enriched in driver mutations, target amino acids of Cys and Gln feature more prominently [11, 13]. Mutation to His is a potential route to gain of function pH-sensing at close to neutral pH [39]. Here the high incidence of mutations from Arg, due to underlying DNA mutational signatures [40] are, if anything, depleted for those to His in the instGE10 set.

When somatic mutations are separated for 15 cancer types, using cancer type names from the Human Protein Atlas [32], the prominence of Arg in the instGE10 set remains a common

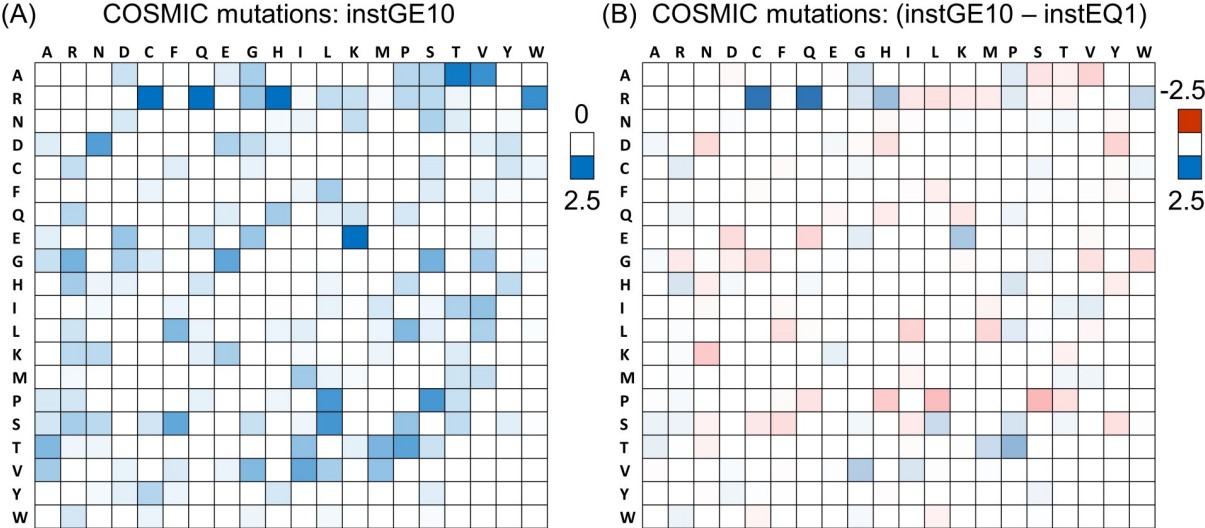

**Fig 1. Mutation matrices across cancer types in COSMIC.** (A) The percentage distribution of all 20 x 19 missense AA mutations in COSMIC, only for mutations with instances $\geq 10$. Wild type AA is listed at the left, and target AA across the top. Summation is to 100% over the entire heat map, and the white to blue colour scale is indicated to the right. (B) Percentage distribution for the instances = 1 subset is subtracted from that for instances $\geq 10$, therefore with sum to 0% over the heat map, and with colour scale (red-white-blue) shown to the right.

**Fig 2. Mutation distributions by cancer type.** (A) Percentage distributions by cancer type, for instGE10 data, with the 20 wild type AAs listed across the top, and each row (cancer type) summing to 100%. Percentage scale (white to blue) is shown to the right. (B) Skin cancer, instGE10 data with the 5 most prominent wild type AAs in panel (A) listed on the left. Distributions of mutation to target AAs (listed across the top) are shown for instGE10 data, percentage scale (white to blue) shown to the right. (C) The distributions of panel (B) are differenced with instEQ1 data, with scale of percentages (red-white-blue) on the right.

feature despite the variation in DNA mutational signatures (Fig 2A). Skin cancer has the most feature-rich set of mutated AAs (Fig 2A), related to the mutational mechanisms resulting from exposure ultra violet light [41], with Glu, Gly, Pro, Ser adding to Arg as heavily mutated. The prevalence of mutations from these 5 AAs, in the instGE10 set for skin cancer (Fig 2B), shows similarity with that across all cancers (Fig 1A), with Arg mutation to Cys/Gln and to a lesser degree, His, as well as Glu to Lys mutation. While the prevalence of mutations from Pro and Ser are a specific feature of skin cancer in the instGE10 set (Fig 2A), enriched destination AAs of Leu (from Pro and Ser), and Ser (from Pro) are also common to the overall cancer landscape for the instGE10 set (Fig 1A). Skin cancer has well-characterised mutational signatures [42] that underpin AA mutations. However the detail of AA mutation profiles shifts for the instGE10 set, compared with instEQ1, likely reflecting the importance of adaptation to tumour conditions and growth (Fig 2C), and similar to the more general results over all cancers (Fig 1B). Of particular note are the persistence of mutations from Arg (to Cys, Gln, His), as the number of COSMIC instances increases. Driver missense mutations impact proteins through loss-of-function (LoF) or gain-of-function (GoF) effects, including mutations from Arg (e.g. the metabolic switch R132H in IDH1, [21]). To what extent the prominence of mutations from Arg incorporates Arg-specific functional effects remains an open question, and the reason that potential pH-sensing for Arg to His mutations has been raised [20].

### 3.2 Predicted ΔpKa and SASA values at Arg and His sites, compared between low and higher instance mutations

Using AlphaFold2 models [23], predictions of ΔpKa values for Arg and His were made with pkcalc [25], together with SASA calculations. Missense mutations in COSMIC for the most prevalent 4 target AAs for each of Arg (Cys, Gln, His, Trp) and His (Tyr, Pro, Gln, Arg) were included, with the data further divided into sets of mutations at just a single instance recorded, and those at $\geq 10$ instances. Overall, sites of mutation from Arg have moderately positive predicted ΔpKas, indicative of structural stabilisation, with very little difference between target AAs, or between 1 and $\geq 10$ instance subsets (Fig 3A). For SASA, there is also little difference between target AAs, but now a systematic reduction in average SASA for $\geq 10$ instance subsets

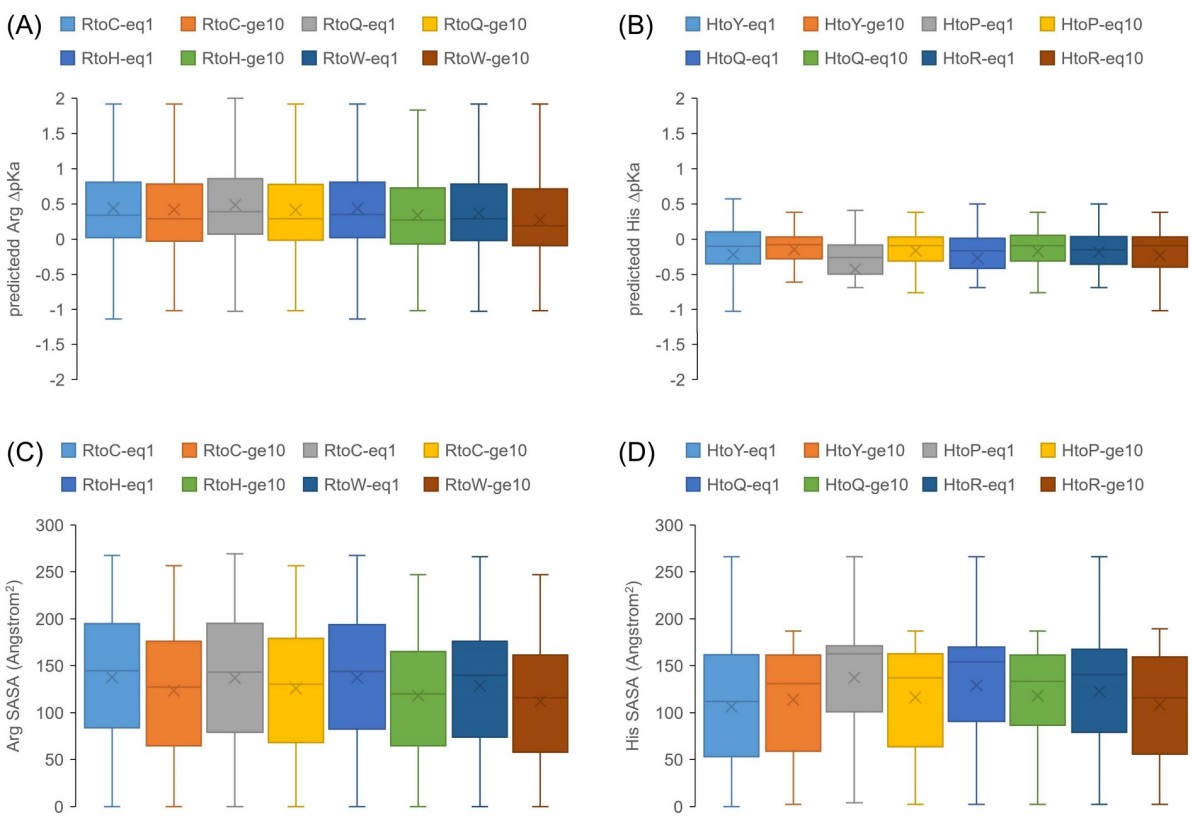

**Fig 3. Structure-based SASA and ΔpKa calculations for mutations from Arg and His.** The box and whisker plots show quartiles, median (central line), and mean (cross). (A) Distributions of predicted Arg ΔpKas (pkcalc) are shown for instEQ1 and instGE10 mutation data, further divided into the 4 most common target AAs for Arg mutation. Limiting thresholds of +/- 3 are applied for ΔpKa values. (B) Equivalent format ΔpKa data to panel (A) are shown for mutation from His, to its most common target AAs. (C) The ΔpKa data for Arg mutations in panel (A) are replaced with SASA, for the same subsets. (D) His mutation SASA is shown, using the same datasets as for His ΔpKas in panel (B).

versus 1 instance subsets with the same target AA (Fig 3C). Notably, there is a fraction of all Arg mutation sites that have very low SASA. These do not map to very low predicted ΔpKas (which would result for uncompensated dehydration), rather they are cases of Arg involved in specific structural stabilisation, often through hydrogen bonding to main chain carbonyl groups. With regard to His target AA as compared with Cys, Gln, Trp, there is no clear difference of SASA or predicted ΔpKa values, and thus no support for a general separation of Arg to His from other target AA mutations, at least for these properties.

Focussing on mutations from His, the most evident features for predicted ΔpKas are low ranges and overall a small negative value, across all displayed target AAs, and in either the 1 instance or $\geq$ 10 instance mutation sets (Fig 3B). This reflects the majority of cases in which (at physiological cytoplasmic pH) His will be neutral and not involved in charge networks. Nevertheless, as for Arg, there is a fraction of His mutation sites with very low SASA (Fig 3D), which are likely to be contributing to the overall slightly negative ΔpKas. Such sites will be buried with little influence of a predicted negative ΔpKa on structural stability, so long as ambient pH is greater than normal His sidechain pKa (6.3 in pkcalc).

The overall picture is that no clear predicted differences of ΔpKa or SASA properties are evident in respect of mutations from Arg to His relative to other target AAs, or His to what may be considered a possible buried sterically similar AA (Tyr), relative to other target AAs. It

is possible, as indicated in the low SASA fractions (Fig 3C and 3D), that certain subsets may underpin pH-dependence, requiring consideration of a variety of factors, including AA burial and ambient pH of subcellular location. For this study the focus is on His as a potential mediator of adaptation to the altered pH micro-environment of tumours [20], but other AAs can also mediate pH-dependence at neutral or mild acidic pH, in particular Asp and Glu [30]. In order to assess whether differences between lower and higher instance COSMIC missense mutation sites are apparent for other amino acids with ionisable sidechains, the same calculations made for Arg and His in Fig 3 were applied to all Asp, Glu, and Lys mutations sites in COSMIC, using AlphaFold2 models (S1 Fig). Predicted ΔpKas for Asp and Glu tend to be negative, and are largely positive for Lys (S1 Fig panel A), in both cases indicating overall stabilisation of the ionised state in the network of charge interactions. Solvent accessibilities are overall larger for Lys than for Asp and Glu, as expected (S1 Fig panel B). In a repeat of the overall result for Arg and His (Fig 3), there are no clear predicted differences of ΔpKa or SASA properties for the COSMIC instance of 1 category compared with that of ≥10 instances. The low SASA tail for Asp, Glu, and Lys residues, as for Arg and His, could be indicative of sites that are relevant for pH-dependence.

### 3.3 Filtering for somatic mutations to His that could mediate pH-dependence

Using AlphaFold2 structures to calculate physical properties, this study omits environments (in particular burial from solvent) that are only realised in homomeric protein or heteromeric (protein or other partner) interactions. Histidine is a prime AA for mediating pH-dependence, especially when buried and with ambient pH at or lower than the normal sidechain pKa. It is therefore used to filter for the potential introduction of pH-sensing through mutation. The procedure is not intended to be exhaustive, rather, given the lack of general findings across classes of mutations, it is an attempt to identify specific examples. An additional feature is whether GoF mutations, as anticipated for generating a pH-sensing His, can be distinguished from LoF mutations. The extent to which a mutation site has single or multiple target AAs in COSMIC is denoted as percentage specificity, with 100% indicating that all mutations are to a single target AA. To restrict analysis to a set of mutations most likely to be drivers, the top 1,000 COSMIC mutations, in terms of numbers of instances in COSMIC (down to 39), were filtered for His target AA mutations (57 for any SASA value in the protomer model). Literature analysis revealed 11 of these as known to be GoF, with 18 LoF mutations. Percentage specificity for the GoF mutations is on average higher than that for the LoF mutations (Fig 4, Mann-Whitney p = 0.0193), suggesting that this feature could contribute to a filter that distinguishes GoF from LoF mutations [43].

The set of mutations to His in COSMIC were studied for potential GoF pH-sensing, with filtering for burial (SASA ≤ 20 Å$^2$ in the protomer model), giving 55 sites down to ≥ 10

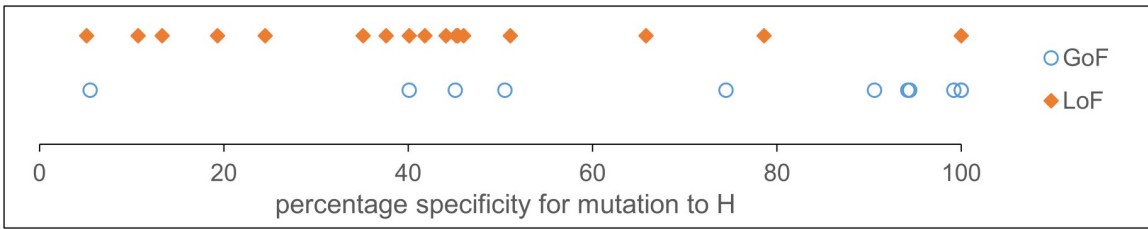

**Fig 4. High instance GoF and LoF mutations compared for His target AA.** Percentage specificity (see text) for mutations to His within the top 1,000 of all mutations, ranked by instances, in COSMIC.

instances. Following the analysis of GoF *versus* LoF mutations, a filter of percentage specificity of mutation at a site (to His) of $\geq 50\%$ was applied, with reduction to 41 sites. Subcellular location was used to further restrict the data, concentrating on cases where a low environmental pH would yield a destabilising influence of the buried His, when loss of hydration is unmatched by protein charge interactions. The resulting 9 cases (Table 1) include some where an alternate explanation to potential pH-dependence is already known or seems likely, and those where a role for introduced pH-dependence is hypothesised. Those 9 are supplemented by a tenth (potassium channel) case, where the site is accessible in the protomer model, but substantially buried in the functional tetramer. Not included in the 10 cases are the two most recurring mutations to His in COSMIC, both well-characterised, IDH1-R132H and p53-R175H. The IDH1 mutation leads to a metabolic switch that is proposed to promote tumour growth [44], also with the suggestion that the mutant enzyme activity could sense environmental pH through a pH-dependent heterodimerisation of IDH1-R132H and IDH1-WT enzymes [21]. R175H of p53 has been classified, within the large set of important p53 mutations, as reducing DNA binding affinity through destabilisation of structure around a zinc binding site [45].

A feature of COSMIC and related databases is their expansion with available cancer genome sequence data. In order to assess the potential influence of such expansion on the current analysis, a comparison was made between the data available on the COSMIC web site in October 2024 and that used generally in the current study (from February 2023). Of the 10 mutations listed in Table 1, for 4 the number of recorded instances remains the same, for 2 mutations instances increase by 1, and increases for the other 4 mutations are 3, 5, 8, and 9 instances. There are 6 mutations in the February 2023 dataset that would pass the burial filter applied for Table 1, but lie just 1 instance below the applied threshold of 10. Of these, 5 remain at 9 instances and 1 increases to 16 instances in October 2024. This comparison of data snapshots 20 months apart shows that whilst numbers will necessarily change over time, there is also a high degree of consistency between the two datasets.

**Table 1. Mutations to histidine: 10 examples with COSMIC instances $\geq 10$.**

| mutn | inst | protein | UniProt ID | mutn spec | pka-pkcalc | pka-PROPKA | location | SASA | PP2 | SIFT | AM | Fig |
|---|---|---|---|---|---|---|---|---|---|---|---|---|
| **L702H** | 85 | ANDR | P10275 | 100 | 2.32 | 3.77 | cyto-nucl | 1.9 | **1** | **0** | **1** | |
| **Q638H** | 11 | PCDHB16 | Q9NRJ7 | 100 | < 1 | 5.05 | TM-EC | 1.7 | 0 | 0.89 | 0.05 | |
| **D268H** | 14 | PCDHGB4 | Q9UN71 | 88 | 10.57 | 7.07 | TM-EC | 16.2 | **1** | **0** | **0.95** | 5A |
| **R191H** | 19 | TINAG | Q9UJW2 | 54 | 2.77 | 3.33 | secr-ECM | 8.4 | 0.84 | 0.09 | 0.27 | |
| **R350H** | 16 | TMEM168 | Q9H0V1 | 80 | 4.04 | 5.28 | TM-nucl | 19 | **1** | **0** | **0.94** | |
| **Q793H** | 11 | NDST1 | P52848 | 100 | < 1 | 4.57 | Golgi | 1 | **0.99** | **0.01** | **0.98** | 5B |
| **R149H** | 16 | MGAT4C | Q9UBM8 | 67 | < 1 | 4.95 | Golgi | 13.7 | **0.97** | 0.16 | 0.24 | |
| **Y195H** | 17 | HLAC | P10321 | 94 | < 1 | 3.94 | TM-ER-EC | 1.6 | 0.16 | 0.09 | 0.22 | 5C |
| **Y115H** | 41 | AQP7 | O14520 | 100 | < 1 | 4.73 | TM-EC | 6.6 | **1** | **0** | **0.68** | 5D |
| **Q192H** | 32 | KCNJ12 | Q14500 | 100 | 5.59 | 5.96 | TM-cyto | 129.2 | 0 | **0.01** | **0.59** | |

mutn spec is mutation specificity; location includes cyto/cytoplasm, TM/trans-membrane, EC/extra-cellular, secr/secreted, ECM/extra-cellular matrix, nucl/nucleus, ER/endoplasmic reticulum; SASA is calculated for the unmutated amino acid (Å$^2$); PP2 is PolyPhen-2 prediction of mutation effect from 0 (benign) to 1 (damaging); For SIFT predictions, values < 0.05 are deleterious and > 0.05 are tolerated; AlphaMissense (AM) predictions of mutation effects are either benign (< 0.5) or pathogenic (> 0.5). For PP2, SIFT, and AM methods, results for mutations predicted to be deleterious are shown in bold and underlined.

### 3.4 Prominent mutations to His in COSMIC, at buried sites

**3.4.1 L702H of androgen receptor.** The 10 mutations listed in Table 1 are contained within proteins that are either located in mild acidic pH environments (e.g. Golgi lumen), neutral pH environments (e.g. cytoplasm), or on the extracellular side of the plasma membrane (neutral or mildly acidic, depending on the tumour environment). Mutations were modelled within the AlphaFold2 protomer using SwissModel [38], and pKa predictions made with pkcalc and PROPKA. Predictions of benign or deleterious to protein function are mostly consistent between PolyPhen-2, SIFT, and AlphaMissense methods for these mutations, indicated by highlighting of deleterious predictions in Table 1. For 5 mutations within the group of 10, all methods predict deleterious, for 3 mutations all methods predict benign, and the remaining 2 mutations have mixed predictions. It is likely that the specific property of introduced pH-dependence is not fully accounted for in the standard mutation effect prediction methods. For example, one of the 4 mutations that is represented in Fig 5 (HLA-C Y195H) is consistently

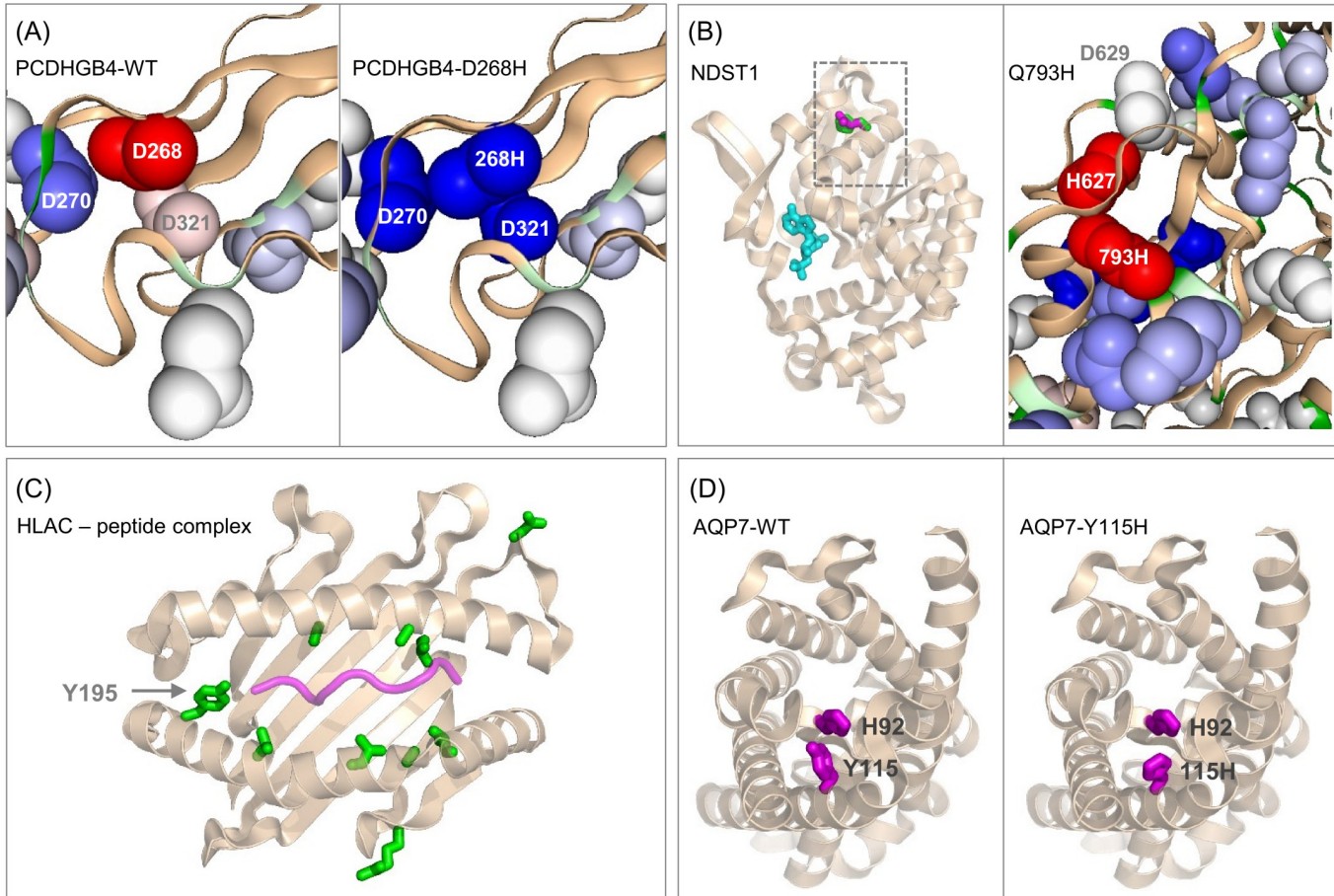

**Fig 5. Molecular environments of selected mutations to His.** (A) PCDHGB4-D268H: left is a protein-sol image of predicted pKas around the wild type calcium binding site (in the absence of calcium ion), with colour scale by destabilisation (red) through ΔpKa close to zero (white) and stabilisation (red). Right is the equivalent plot for the D268H mutant, with a switch from destabilising at 268 to stabilising. (B) NDST1-Q793H: left is the sulfotransferase domain from a structure of the bifunctional Golgi enzyme (8ccy) [50], with ADP (cyan) marking the active site. Q793 is indicated with a rectangle, and that region expanded on the right, with a protein-sol pKa prediction showing destabilised Q793H and neighbouring H627. (C) HLAC-Y195H: mutation sites (instances ≥ 17 in COSMIC, green) are displayed in the peptide binding domain of structure 6pag [51], with bound peptide shown in purple. Mutations other than Y195H are S48A, L171W, T187M/T187P, A97T, L180R, A176V, L119I, all adjacent to the peptide binding groove, and R155S, D114A in this domain but not adjacent to the peptide binding groove. (D) AQP7-Y115H: left shows wild type Y115 in the AlphaFold2 protomer model, and right the modelled Y115H mutation. In both panels the neighbouring residue H92 is shown.

predicted as benign, but it is not clear how these methods take into account factors such as differential pH of subcellular/extracellular location, altered extracellular pH in cancer, or coupling between the mutation site and ligand binding.

The first example, L702H of androgen receptor (ANDR), is part of the neutral pH subset. This mutation is documented to alter the affinity for some natural ligands, and also for drugs that are targeted at the oncogene ANDR to down-regulate its transcriptional activity in tumour growth [46]. Mutations at relatively buried sites and adjacent to ligand binding pockets can affect processes that impact on tumorigenesis in both enzymes and non-enzymes.

**3.4.2 Protocadherins.** Two of the 10 sites (Table 1) are in protocadherins, a large group of proteins involved in cell-cell adhesion [47]. In PCDHGB4, D268 is part of a cluster of 3 Asp that contribute to a calcium binding site, referencing a mouse PCDHGB4 structure (6e6b), with mouse D238 equivalent to human D268 [48]. Using the pka tool on the protein-sol server [25], a web implementation of pkcalc, pka predictions were made for PCDHGB4-WT and PCDHGB4-D268H, in both cases in the absence of calcium (Fig 5A). Interestingly, destabilisation for parts of the Asp cluster, anticipated in the absence of calcium ion (due to repulsion between the adjacent carboxylate groups), is replaced with stabilising predicted pKa changes for the D268H mutant (with calculated pKa greater than the normal His sidechain pKa value of 6.3). The histidine is predicted to be positively-charged at neutral pH (pkcalc), networking favourably with the remaining Asp sidechains in the calcium binding site. Whether this predicted electrostatic stabilisation, or perhaps an introduction of His-mediated pH-dependence, plays a role in tumour growth is unknown. The other protocadherin mutation in Table 1 is PCDHB16-Q638H. Again a mouse structure is available (5szq), with mouse Q612 equivalent to human Q638 [49]. In this case Q638 is not part of, or adjacent to, a calcium binding cluster of AAs. Since the site is buried and the predicted ΔpKa is negative, incorporation of His could lead to pH-dependence if the EC pH is depressed close to the normal His sidechain pKa (6.3).

**3.4.3 R191H in TINAG.** TINAG is a secreted protein that is a constituent of basement membranes. It has been implicated in cancer, with expression levels correlated with survival of patients with kidney renal clear cell carcinoma [52]. R191H is the most common TINAG missense mutation in COSMIC. Whereas the buried R191 is able to salt-bridge with D236, and as a result is a predicted stabilising influence on structure, R191H in the model is not able to make the same salt-bridge, losing some stability. Further destabilisation would occur should the environmental pH fall to around the normal His sidechain pKa. No specific information on the effect of TINAG-R191H is available.

**3.4.4 R350H in TMEM168.** A multi-pass TM protein in the nuclear membrane, TMEM168 has a reported association with glioblastoma multiforme (GBM) [53], with no indication of mechanism for R350H, its most numerous mutation in COSMIC. This mutation lies adjacent to a modelled TM segment. As for TINAG-R191H, it is predicted to mutate from a stabilising network of Arg interactions (in this case with backbone carbonyls), to a largely buried His. In this case though the loss of stabilising Arg is unlikely to be supplemented by His-mediated pH-dependence, since the environment pH is well above the free His sidechain pKa.

**3.4.5 Q793H in NDST1.** The single pass TM protein NDST1 is a Golgi luminal bifunctional enzyme involved in heparan sulphate modification. Q793 lies in a flanking domain adjacent to the active site carrying the sulfotransferase activity (8ccy) [50], buried and making hydrogen-bonds with two backbone carbonyl groups. A modelled Q793H is unable to make these interactions, with a predicted acidic pKa (Fig 5B). Destabilisation relative to wild type is predicted to occur both through the loss of interaction to neighbouring main-chain and from a reduced stability around the introduced His at the acidic Golgi pH of 6.0 to 6.7 [54]. Of interest is a neighbouring and buried His (H627), predicted to have a negative ΔpKa in wild type AlphaFold2 model (pkcalc and PROPKA) and mutated enzyme (pkcalc), as well as in the

cryo-EM structure (8ccy, pkcalc), despite the adjacent Asp (D629). The Q793H mutation could be adding to existing pH-dependence mediated by H627, or the interaction of H627 with D629 could be more stabilising than predicted, so that H627 would not be a source of structural instability at Golgi pH. Whatever the case, any His effect will depend on the precise value of Golgi pH relative to the free His sidechain pKa (6.3) and could, in principle, act as a protein stability sensor of Golgi pH. It has been reported that a microRNA upregulated in GBM tissues targets and negatively regulates NDST1 [55], consistent with the hypothesis that complementary protein structure destabilisation could play a similar role.

**3.4.6 R149H in MGAT4C.** MGAT4C is also a single pass TM protein, here with glycosyl-transferase activity and R149 located in the Golgi lumen. R149 and D150 interact in the Alpha-Fold2 model. R149H (with low predicted pKa, Table 1) could act as a pH sensor if Golgi pH is comparable to the free histidine sidechain pKa. If the mutation is detrimental to enzyme activity, this would be consistent with reported increased GBM tumour growth when MGAT4C is under-expressed [56].

**3.4.7 Y195H in HLA-C.** HLA-C Y195 is buried and located adjacent to the peptide binding groove (with reference to a reported complex structure, 6pag [51]). While His is sterically similar to Tyr, without specific protein charge solvation it has a low predicted pKa (Table 1), and could be a source of structural instability where environmental pH falls to around 6.3. This is perhaps attainable in the acidic extracellular environment of tumours, but is also possible in the exocytotic pathway [57] that transports peptide-loaded HLA-I complexes from the ER to the cell surface. Additionally, of the 9 mutations in COSMIC with equal or greater number of instances to Y195H, that are located in the peptide binding domain of HLA-C, 7 are adjacent to the peptide binding groove and (as suggested for Y195H) could therefore affect binding affinity (Fig 5C). This mechanism could contribute to the inhibition of immune surveillance that is a feature of tumorigenesis [58].

**3.4.8 Y115H in AQP7.** Aquaporin-7 (AQP-7) assembles in the cell membrane as a homo-tetramer. Each protomer contains a pore for water and glycerol. Structural data reveal that Y115 is adjacent to the pore, as is a neighbouring AA, H92 (6qzi, [59]). This pair of AAs is conserved in AQP-10 (Y103 and H80, [60]), where H80 is proposed to mediate pH-dependence with protonation at pH 5. Predicted pKas of H80 in the AlphaFold2 protomer model of AQP-10 are 3.4 (pkcalc) and 3.9 (PROPKA), noting again that His pH-dependence depends on pH falling below the normal His sidechain pKa (6.3), and thereby favouring a less buried environment. An argument is made that since H92 of AQP-7 replicates H80 of AQP-10, and that AQP-7 does not exhibit pH-dependence of pore activity (between pH 6 and 8), other conformational effects are involved [59]. Indeed, predicted pKas for AQP-7 H92 in the wild type AlphaFold2 monomer are still low at 3.3 (pkcalc) and 5.09 (PROPKA), so that predicted pH-dependent instability around normal His sidechain pKa is insufficient to alter conformation, given the lack of observed pH-dependent function. A factor that could change the conformational balance is introduction of a further buried His (Y115H mutation). It is possible that Y115H introduces a pH-dependent pore function to AQP-7, similar to that of AQP-10 (Fig 5D).

**3.4.9 Q192H in KCNJ12.** In a further example, Q192H of KCNJ12, the Kir2.2 inwardly rectifying potassium channel, is accessible within the AlphaFold2 protomer model (a difference to the other 9 sites in Table 1), with both pkcalc and PROPKA predicting a pKa slightly depressed to mild acidic pH. Since this AA is located on the cytoplasmic side of the membrane, it is unlikely to encounter an environmental pH much below neutral. However, it also lies next to the inositol 4,5-bisphosphate group of channel regulator PIP2, which contributes to control of channel opening [61]. Additionally the Q192 site is now much less accessible due to the adjacent PIP2, and a neighbouring protomer in the channel tetramer. Interaction of the

Q192H sidechain with neighbouring phosphate groups could elevate its pKa, and introduce a pH-dependent element to channel opening at a pH close to neutral. Notably, other sources of electrostatic modulation on the cytoplasmic face of Kir2.2 activity have been discovered [62].

## 3.5 Context on filtered His mutations

In a search for sites where somatic mutations may underpin tumour adaptation to pH, several filters have been applied (introduction of buried His in AlphaFold protomer models, instances in COSMIC of at least 10, consideration of the environmental pH). These are restrictive, noting for example the role of Asp and Glu in pH-dependent processes [30], yielding a relatively small number of sites. There are systems with histidine mutations, not revealed in the current work, where a pH-dependence of molecular function has been established and potential association to cancer suggested. These include IDH1-R132H ([63], already discussed; β-catenin H36P and other target AAs [22], (an accessible site involved in protein-protein interactions); and several systems where the summed instances in COSMIC is just one, RasGRP1-H212 [64], FAK1-H58 [65], a 4 His cluster in NHE1 [66]. These characterised systems can be used to assess the effectiveness of predictions of pH-dependence, although they are not in Table 1. The indicated sites, (Arg in IDH1, His in RasGRP1 and FAK1), are buried ($< 20$ Å$^2$ SASA) with large predicted pKa changes, and would therefore be flagged as of interest in respect of pH-dependence. The loop carrying the 4 His cluster of NHE1 is exposed to solvent in the Alpha-Fold model of NHE1, but largely buried at the interface with obligate binding partner CHP1 [67], demonstrating one way in which the current computational pipeline can fail, where pH-dependence is only manifested when burial occurs at an interface between molecules. The H36P β-catenin mutation falls into a similar category, although with the added complexity that H36 is in an intrinsically disordered region, in the absence of a binding partner. The advent of AlphaFold models protomers is in itself a very significant step forward for the type of analysis reported here, allowing complete human proteome coverage. Greater coverage of the human biomolecular interactome, through experimental and modelling [68] methods, will yield further insight.

Referring to the experimental characterisations of pH-dependence, it appears that for some of these the mechanisms of pH-dependence are not sufficiently susceptible to modulation by somatic mutation or sufficiently coupled to tumour growth, to appear at high incidence in the COSMIC database. Of note is that transcriptional and other changes that alter protein levels are common in tumorigenesis. For example, it may be more simple to enhance sodium / proton antiporting activity of NHE1, and response to acidosis [69], through transporter levels than through modulation of intrinsic activity.

## 3.6 Net mutation from arginine and to lysine is evident in COSMIC

The percentage of Arg and Lys sites that are in COSMIC both correlate with the overall percentage of sites mutated in that protein. Whereas the percentage of Arg sites mutated exceeds the average number, for the majority of proteins, the inverse is generally the case for mutation from Lys (Fig 6). This effect is evident for datasets covering all COSMIC mutations (Fig 6A), and including only those sites with at least 10 instances of a mutation (Fig 6B). Emphasis of mutations from Arg over those from Lys is complemented by an inverse effect for mutations to Arg or Lys, including the prevalent Glu to Lys mutation (not shown). It is apparent that whatever the DNA mutation mechanisms behind amino acid changes, they result in an overall shift from Arg to Lys.

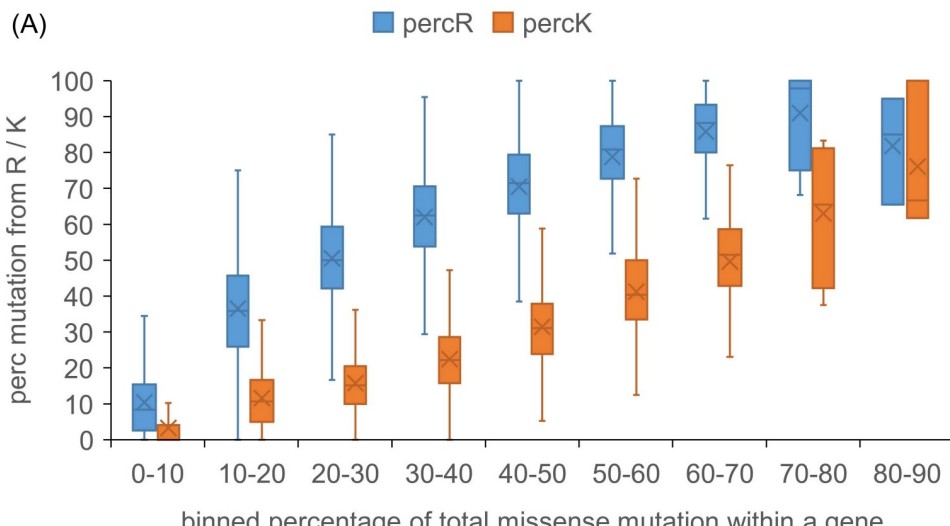

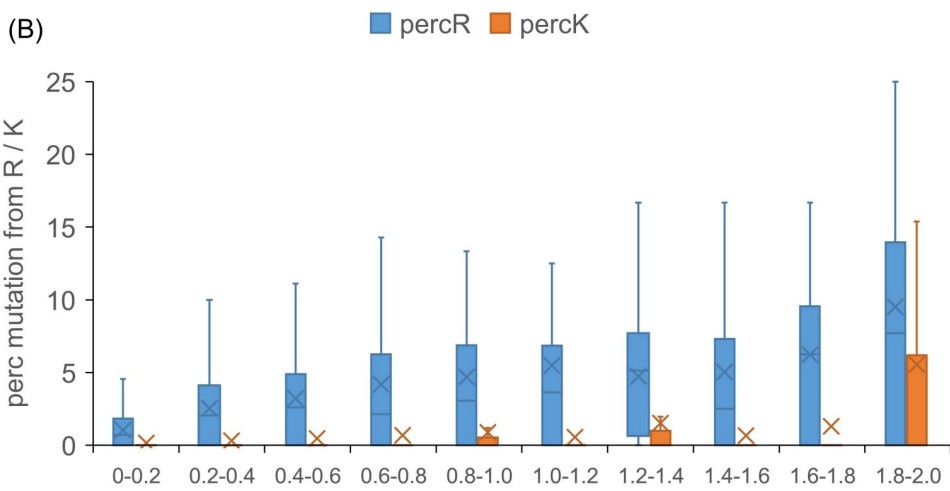

**Fig 6. Extent of missense mutation from Arg and Lys in COSMIC compared with overall mutation.** Underlying data are the overall percentages of locations in each gene that have mutations recorded in COSMIC, and also those percentages when considering just wild type Arg and Lys sites. The overall percentage data are binned (horizontal axis) and displayed with the Arg or Lys data on the vertical axis. (A) Analysis for all missense mutations in COSMIC. (B) Analysis for sites with ≥ 10 instances in COSMIC.

### 3.7 Lysine to arginine balance is a feature in studies of protein hydrophobicity and solubility

Given that the incidence of Arg to His mutations does not appear to be systematically linked to modulation of pH-dependence, it is reasonable to ask what other consequences there may be for mutation from Arg. Although this is widespread in different cancer types, and results from specific mutational mechanisms to generate DNA changes, there remains a question of how advantage may accrue for tumour growth, as seen for skin cancer on the background of a particular mutational signature (Fig 2C). One suggestion has been that mutation to cysteine could counter production of reactive oxygen species during tumour growth [13]. Another

possibility, discussed here, is that a rebalancing of Arg and Lys could be beneficial for maintaining cellular proteostasis in the rapid growth of tumours.

Several studies have pointed to the differentiated properties of Lys and Arg in scales of hydrophobicity and solubility. Locations of AAs in protein structures have been used to derive a stickiness scale, in which Lys is the least sticky AA with Arg lying close to the middle of the 20 AAs [70]. Combining this scale with abundance data, it was found that more abundant proteins have less sticky surfaces. Divergence between results from a large-scale study of AA contributions to protein stability, and evolutionary AA usage, was used to construct a scale representing the influence of non-stability factors on AA conservation [71]. It was suggested that solubility is a major contributor, and Lys (substantial divergence) is well separated from Arg (little divergence). Comparing Lys and Arg content with solubility data revealed that Lys is enriched relative to Arg in many of the more soluble proteins [72]. This observation contributes to the protein-sol solubility prediction tool [73]. A scale based on structural flexibility has also been used to make a model for protein solubility, with Lys again separated from Arg, grouping with Asp and Glu at the more soluble end of the spectrum [74].

The observation that Lys/Arg balance is a factor in determining protein solubility, and that it is modulated in the dataset of cancer somatic mutations, is of interest when considering protein homeostasis (proteostasis) in cancer. Links have been made between the role of proteostasis in ageing and in cancer [75], and the pursuit of cancer therapies that target proteostatic factors has been reviewed [76]. Examples of the latter include targeting of HSP90 [77], and HSF1 [78]. Further, it has been shown that a major part of the *Caenorhabditis elegans* proteome is close to intrinsic solubility limits [79]. If this fine solubility balance extends to tumour cells in humans, where altered protein expression is common, then it might be expected that somatic mutations that modulate some aspect of solubility would occur.

### 3.8 Arginine stands out in variation with COSMIC instances, and conservation

In order to catalogue the over-representation of arginine somatic mutation (summed over target amino acids), both the absolute numbers (Fig 7A), and distributions (Fig 7B), for wild type AAs are calculated, for various ranges of instances in COSMIC. Several amino acids are close in terms of highest numbers of single instance sites (Fig 7A). Notably, all the AAs with one instance numbers comparable or larger than Arg (Ala, Glu, Gly, Leu, Pro, Ser) have higher AA composition values in the human proteome. That the proportion of mutations at arginine (Fig 7B) increases above one instance, demonstrates the higher propensity for mutation of Arg. This proportion, for Arg, falls for the category of $\geq 10$ instances, but remains larger than for other AAs. Single instance numbers are used, along with AA composition, to estimate an overall background probability of mutation for each amino acid (see Materials and methods). Calculated probabilities are then applied to estimate the expected number of mutations for each wild type AA, at different numbers of instances, in the non-driver case. Finally, the modelled distributions are subtracted from the observed distributions (Fig 7B) to yield the differences (Fig 7C). Many AAs show moderate increases in their share of the overall distribution (in actual versus modelled), as the instances range rises to $\geq 10$, at the expense of a large drop for Arg. Despite this fall for Arg, relative to a model without adaptation, it remains the largest contributor to mutation sites for instances $\geq 10$ (Fig 7B). Presumably the relative fall in Arg mutation occurs as adaptation plays a more important role for sites with higher instances. While this effect is conveniently displayed in terms of the percentage distributions across AAs, it is instructive to note the absolute numbers. For the $\geq 10$ instance subset, there are 3,008 Arg sites in a total of 12,121 in COSMIC, compared with 560 Arg from a total of 611 (background

(A)

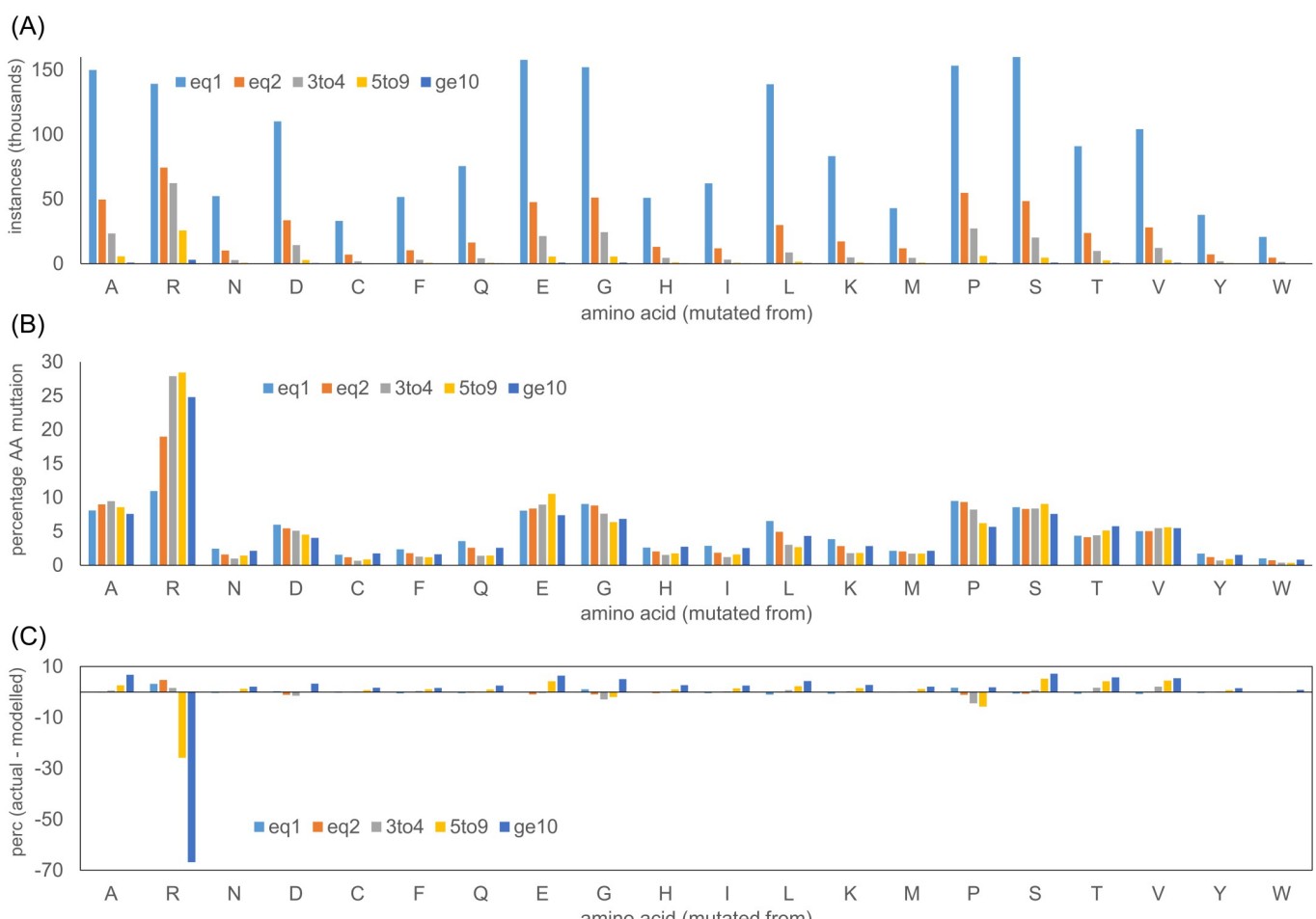

**Fig 7. The distribution of missense mutation sites in COSMIC.** (A) Numbers of missense mutation sites for the 20 AAs are shown for different ranges of instances recorded in COSMIC. (B) Percentage distributions of missense mutations for the 20 AAs. (C) Percentage distributions, modelled from the frequency of single instance mutations (see Materials and methods), are subtracted from those observed in COSMIC, to estimate the effect of adaptation at higher instances.

model), noting that these calculations are made with instances summed over all target AAs at a site. Despite the simple model employed, differentiation of these numbers (actual to model), likely demonstrates the impact of adaptation as the number of instances increases. Notably, mutation from Arg is dominant in all subsets of instances.

Another property for which Arg behaves differently as the number of instances increases is the Functional Impact Score (FIS), available as pre-calculated values for the human proteome from the Mutation Assessor tool [33]. The basis for FIS and MA scores is multiple sequence alignment, so that it is primarily recording evolutionary conservation. The method performs comparably with others in predicting non-synonymous single nucleotide variant pathogenicity [80, 81]. For most AAs, the MA score and therefore conservation is substantially lower for mutations with higher number of instances (Fig 8). Arg shows the most difference to the overall trend, with relatively small changes in conservation, and no reduction for the $\geq 10$ instances subset. The general shift to lower conservation in the highest instances subset (Fig 8) may reflect a balance shifted towards GoF mutations. Similarly the much smaller change for Arg could indicate a lesser role for GoF mutations. Examples of Arg LoF mutation include

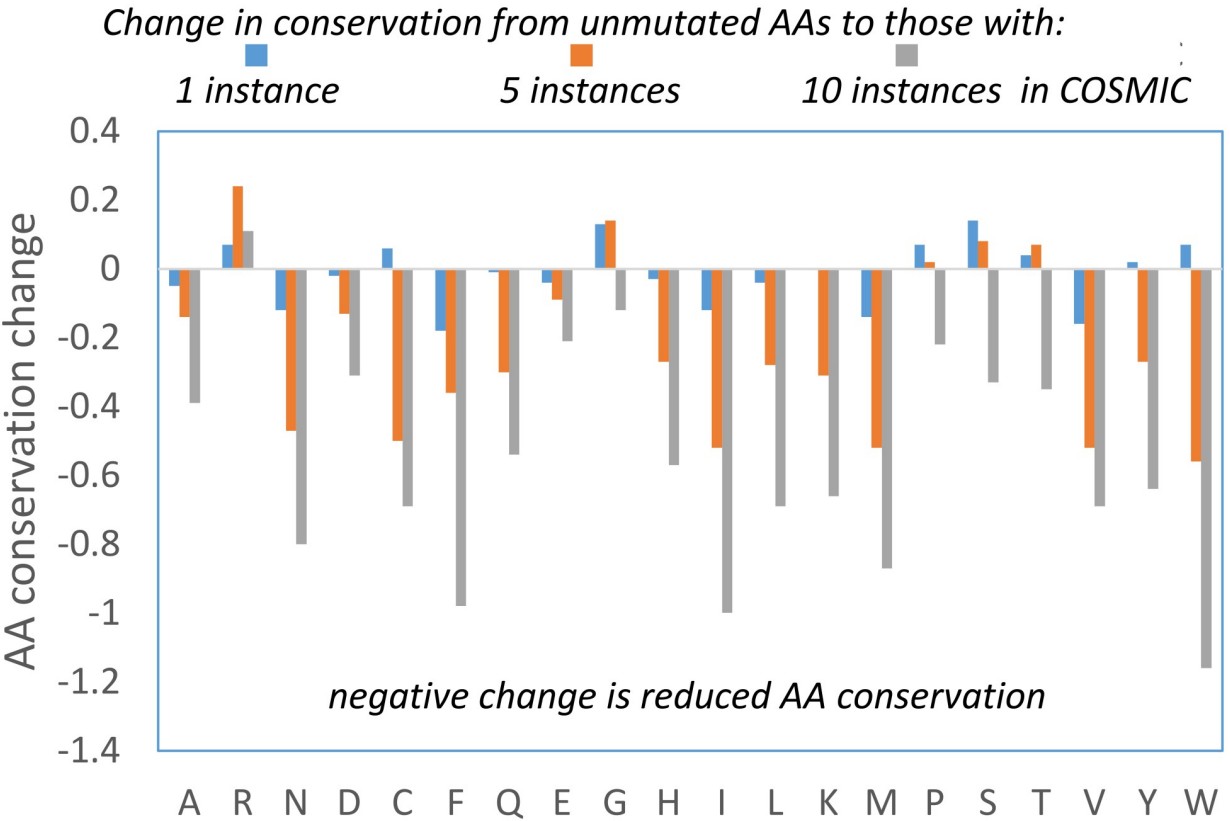

**Fig 8. Analysis of AA conservation with increasing COSMIC instances.** (A) Change in FIS (representing AA conservation), with the AA values averaged over COSMIC instance populations of missense mutations subtracted from the equivalent averages for AAs at sites that do not appear in COSMIC.

protein—DNA interaction sites, for example in p53 [45], and RNA splicing, for example SF3B1 [82]. Even so, conservation for Arg overall changes little across the instances subsets, rather than moving towards greater conservation at higher instances, as would be expected should LoF mutations be dominating in tumour adaptation.

## 3.9 Proteins with mutations from Arg and to Lys, at higher COSMIC instances, are enriched for cell periphery location

To examine the hypothesis that subsets of mutations from Arg, and to Lys, introduced broadly across cancer types, could be combatting proteostasis challenges in cancer cells, gene ontology was examined for sets of proteins with mutations at different numbers of instances in COS-MIC (summed over all target mutations at a site when considering mutation from an AA). Using the Princeton GO Finder tool [34] for the from Arg and to Lys mutations, consistently the cell periphery category in GO component classification is the most significant (for all instances of to Lys mutations, and instances 8, 9, $\geq$10 of from Arg mutations in Fig 9), or close to the most significant (instances 4, 5, 6, 7 of from Arg mutations in Fig 9), category returned. The percentage difference between each set tested and the background human proteome set was calculated for cell periphery (Fig 9). While there is a general increase of mutations for proteins at the cell periphery as number of instances increases (except for the $\geq$ 10 subsets), both from Arg (Fig 9A) and to Lys (Fig 9B) sets are further enriched. Viewed in the context of

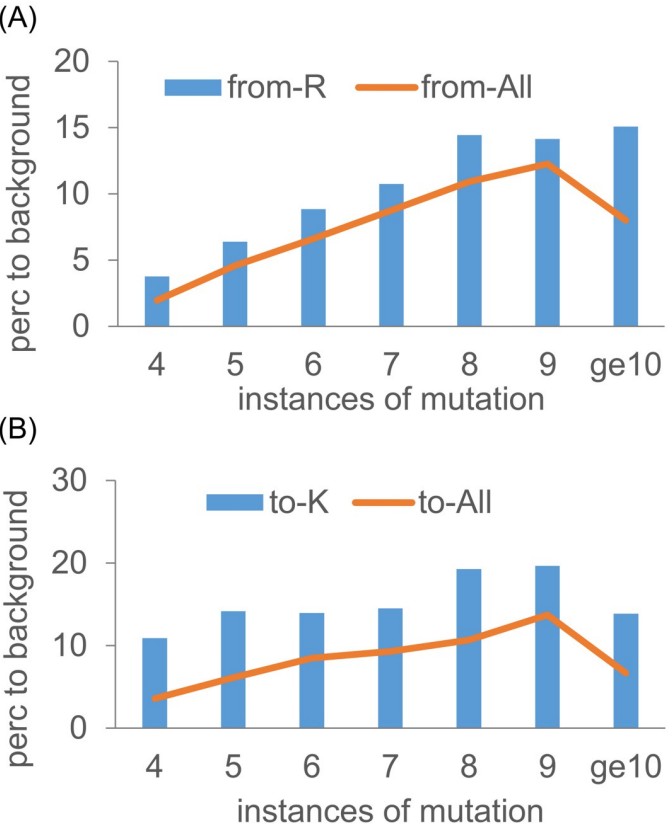

**Fig 9. Cell periphery enrichment of proteins with mutations from Arg and to Lys.** Percentage enrichments for the GO category of cell periphery are plotted for overall mutation populations and from Arg (A), and to Lys (B) mutations. Results are shown for different numbers of mutation instances in COSMIC.

membrane proteins as key cancer hallmarks [83, 84], and the connection between cell surface protein aggregation, membrane protein homeostasis, and the endocytotic pathway [85, 86], it is reasonable to ask whether a subset of somatic mutations could be linked to proteostasis at the cell membrane.

The number of mutations in a tumour varies considerably between cancer types, and within a cancer type [87]. For the missense somatic mutations of the current study, retrieved from the February 2023 COSMIC database, numbers of mutations reported for each unique tumour and sample identifier pair were calculated for a subset of cancer types. At the higher end, in 6,021 sequenced samples, stomach cancers have an average of 62 missense mutations per sample, with 684 samples $\geq$ 100 mutations, and 58 $\geq$ 1,000. For 24,756 skin cancer samples, the average is 34 missense mutations, with 1,119 $\geq$ 100 and 241 $\geq$ 1,000. At the lower end, 22,774 breast cancer samples have an average of 12, with 444 $\geq$ 100 and 14 $\geq$ 1,000. Percentages of all mutations that are either from Arg or to Lys, are (24, 25, 25), and for mutations with $\geq$ 10 instances are (32, 27, 34), for the stomach, skin, and breast cancer, respectively. Given the extent to which proteins are expressed at their solubility limit [79], mutation numbers in some tumour samples may be sufficient to influence proteostasis.

## 4. Conclusions

Mutation from Arg is a dominant feature of missense somatic mutations in cancer (Fig 1A). Arginine depletion [14] overall is related to underlying mutational signatures in DNA, but the

balance of AA mutations varies for higher instances compared with lower instances (Fig 1B). The inventory of missense somatic mutations with greater occurrence in COSMIC is therefore the result of both underlying mutational signature and adaptation for tumour growth, with mutation from Arg prominent throughout (Fig 7). One of the targets of Arg missense mutation is His, which has led to the suggestion that Arg to His mutations may be generating pH-sensing functions that aid adaptation to altered pH in tumour growth [20]. This may be relevant for a subset of Arg to His mutations, but higher occurrence Arg mutations are enriched for mutation to Cys and Gln, relative to His (Fig 1). Investigation more generally for mutations involving His showed no clear signal overall for higher instance sites, compared with lower instance sites, that indicate pH-sensing (Fig 3). Adding further filters, including burial and lower environmental pH, revealed a number of sites at which pH-sensing may play a role in adaptation for tumour growth (Fig 5). Other avenues to study include looking beyond Alpha-Fold protomers, at AAs other than His [30], modelling the coupling between protonation and metal ion binding, and accounting for the pH-dependence of enzyme activity. Computational studies will supplement new experimental tools for measuring the relationships between cancer, pH, transcription and proteome stability [88].

In respect of arginine depletion, missense somatic mutations from Lys are under-represented, opposite to Arg, contributing to a rebalancing of Arg and Lys in cancer genomes (Fig 6). This result is intriguing in the context of Lys and Arg being well separated in a number of hydrophobicity and solubility scales, where Lys is favoured for solubility. It is unknown what level of missense somatic mutations would be necessary to influence proteostasis. Average numbers of missense mutations in cancer genomes are typically in the 10s, but with significant numbers of genomes in the 100s, and some in the 1,000s. Gene ontology analysis suggests that if Arg / Lys rebalancing is a significant factor in proteostasis, then the cell periphery is a candidate location (Fig 9). A recent study has used antibody-mediated cross-linking of cell surface proteins to reveal aggregation-dependent endocytosis and lysosomal degradation as a proteostasis response to altered membrane protein behaviour [85]. An experimental test of any contribution from Arg / Lys rebalancing in membrane proteins could be investigated in a similar way, but substituting altered Arg / Lys content, and perhaps altered membrane protein expression, for antibody-mediated cross-linking. Another possibility is to study the wild type and arginine depleted aqueous phase domains (from the membrane proteins) in solution, or at a surface [89]. A potentially related area is the decline of proteostasis mechanisms in ageing [90]. Since Arg depletion arises, at least in part, from underlying C > T DNA mutations [14], it may also contribute more generally to modulation of proteostasis, although any role of somatic mutations in ageing is not yet clear [91].

Finally, altered expression of proteins in cancer is an important factor, whether in considering proteostasis or in the balance between expression and somatic mutation for mediating adaptation to shifted pHi and pHe in tumours. It may be more effective to tune the level of a protein than to modulate its function through mutation, or introduce new (pH-sensing) functions. This is particularly the case with expression levels of acid-base transporters [5], and also modulation of degradation [92].

## Supporting information

**S1 Fig. Structure-based SASA and ΔpKa calculations for mutated Asp, Glu, and Lys.** The box and whisker plots show quartiles, median (central line), and mean (cross). (A) Distributions of predicted Asp (D), Glu (E), and Lys (K) ΔpKas (pkcalc) are shown for instEQ1 and instGE10 mutation data. Limiting thresholds of +/- 3 are applied for ΔpKa values. (B) The

ΔpKa data for mutations in panel (A) are replaced with SASA, for the same subsets.
(PDF)

## Acknowledgments

The authors thank Sifan Zhang for valuable discussions, and staff at the University of Manchester Computational Shared Facility for facilitating storage and processing of data.

## Author Contributions

**Conceptualization:** Shalaw Sallah, Jim Warwicker.

**Data curation:** Shalaw Sallah.

**Formal analysis:** Shalaw Sallah, Jim Warwicker.

**Funding acquisition:** Jim Warwicker.

**Investigation:** Shalaw Sallah.

**Methodology:** Shalaw Sallah, Jim Warwicker.

**Project administration:** Jim Warwicker.

**Resources:** Shalaw Sallah.

**Software:** Shalaw Sallah, Jim Warwicker.

**Validation:** Shalaw Sallah.

**Visualization:** Shalaw Sallah.

**Writing – original draft:** Jim Warwicker.

**Writing – review & editing:** Shalaw Sallah, Jim Warwicker.

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
