## [Decision Letter · Decision Letter 0]

27 Aug 2024

PONE-D-24-29679Computational investigation of missense somatic mutations in cancer and potential links to pH-dependence and proteostasisPLOS ONE

Dear Dr. Warwicker,

Thank you for submitting your manuscript to PLOS ONE. After careful consideration, we feel that it has merit but does not fully meet PLOS ONE’s publication criteria as it currently stands. Therefore, we invite you to submit a revised version of the manuscript that addresses the points raised during the review process.

We look forward to receiving your revised manuscript.

Kind regards,

Rajesh Kumar Pathak, Ph.D.

Academic Editor

PLOS ONE

Journal Requirements:

2. Please note that PLOS ONE has specific guidelines on code sharing for submissions in which author-generated code underpins the findings in the manuscript. In these cases, we expect all author-generated code to be made available without restrictions upon publication of the work. 

Please review our guidelines at https://journals.plos.org/plosone/s/materials-and-software-sharing#loc-sharing-code and ensure that your code is shared in a way that follows best practice and facilitates reproducibility and reuse.

"This work was supported by UK Biotechnology and Biological Sciences Research Council grant BB/V0065921/1 to JW."

4. Please note that funding information should not appear in the Acknowledgments section or other areas of your manuscript. We will only publish funding information present in the Funding Statement section of the online submission form. Please remove any funding-related text from the manuscript. 

**Additional Editor Comments:**

The reviewers have identified several key areas where the manuscript can be significantly strengthened, including broadening the analysis, enhancing the background information, and providing a more thorough discussion of the limitations of the computational methods used. Addressing these points will help the study make a meaningful contribution to the field.

Reviewers' comments:

Reviewer's Responses to Questions

**Comments to the Author**

1. Is the manuscript technically sound, and do the data support the conclusions?

Reviewer #1: Yes

Reviewer #2: Partly

2. Has the statistical analysis been performed appropriately and rigorously? 

Reviewer #1: Yes

Reviewer #2: Yes

3. Have the authors made all data underlying the findings in their manuscript fully available?

Reviewer #1: Yes

Reviewer #2: Yes

4. Is the manuscript presented in an intelligible fashion and written in standard English?

Reviewer #1: Yes

Reviewer #2: Yes

5. Review Comments to the Author

Reviewer #1: Review Report:

The paper explores the role of somatic mutations, with a particular focus on arginine depletion and pH-sensing mutations, in cancer cell adaptation to acidic environments and their potential impact on tumor progression. The study employs advanced computational tools, including AlphaFold2 and pKa prediction methods, to analyze the structural and functional implications of these somatic mutations.

The paper highlights potential therapeutic opportunities by targeting pH-sensing mechanisms in cancer cells.

Major Comments:

1. The paper primarily focuses on histidine and arginine mutations, neglecting other potentially relevant mutations that could provide a more comprehensive understanding of cancer cell adaptation. Expanding the analysis to include other amino acid mutations that may play a role in pH-dependent processes would offer a more holistic understanding of the mechanisms involved.

2. The introduction and background sections do not provide enough context on the broader landscape of somatic mutations in cancer, limiting the reader's understanding of the study's significance. Enhancing these sections to include a more detailed discussion of the general landscape of somatic mutations in cancer will help readers better appreciate the study's relevance and significance.

3. The paper places too much emphasis on computational data without adequately discussing the limitations and potential inaccuracies of these methods. A more balanced discussion is needed, critically evaluating the limitations of the computational methods used and discussing potential inaccuracies and their impact on the study's conclusions.

Minor Comments:

1. The introduction should be expanded to provide more context on the broader landscape of somatic mutations in cancer, which is crucial for understanding the study's significance.

2. The results are presented clearly, but the discussion does not adequately address the computational methods' limitations. Additionally, the focus on histidine and arginine mutations is too narrow.

The paper provides valuable insights into the role of arginine depletion and pH-sensing mutations in cancer. However, major revisions are required to address the lack of experimental validation, narrow focus, and insufficient contextual background. With these revisions, the study could make a substantial contribution to the field.

Reviewer #2: Comments to the Authors:

The study titled "Computational Investigation of Missense Somatic Mutations in Cancer and Potential Links to pH-Dependence and Proteostasis" by Shalaw Sallah and Jim Warwicker provides a thorough analysis of somatic missense mutations, specifically focusing on the arginine-to-histidine substitution, which may contribute to pH-sensing functions. Within the frequently mutated subset, the authors identified mutations in NDST1, the HLA-C chain of the MHC I complex, and the water channel AQP-7 as potential mediators of pH-dependence in cancer cells. Furthermore, they emphasized that rebalancing the arginine-to-lysine ratio is crucial for maintaining proteostasis in peripheral cellular locations and controlling tumor development. However, I believe there are areas where further improvements could be made.

Major revisions:

1. In the methods section, the authors noted their use of COSMIC database version 97. However, the latest version, 100, has been released, including several new missense mutations. Did the authors consider analyzing these recently added mutations in the current study? If not, it would be beneficial to include this analysis.

2. In Table 1, ten mutations to histidine are listed along with mutation specificity and pKa values. However, the parameters used to analyze the stability of the protein structures are not clearly defined. Please provide the exact SASA (Solvent Accessible Surface Area) values for each mutation. Additionally, include the SIFT, PolyPhen, and CADD (Combined Annotation Dependent Depletion) scores for each mutation to enhance clarity.

3. In Section 3.7, the authors described how the lysine-to-arginine balance could be beneficial for maintaining cellular proteostasis in tumor cells. Have the authors analyzed Lysine/Arginine (Lys/Arg) mutations in previously published studies on centenarians or healthy aging? Does this Lys/Arg balance contribute to healthy aging, potentially acting oppositely to the mechanism observed in cancer cells?

Minor Revisions:

4. The authors discussed the significance of the arginine-to-histidine mutation in cancer cells, highlighting its role in gaining pH-sensing function. They also referenced previously published studies explaining the mechanism behind this mutation. However, are there any other significant functions gained by this arginine-to-histidine mutation specifically in cancer cells beyond pH-sensing? If so, please elaborate on these in the introduction.

5. In Section 3.4.2, the authors discussed the stabilization of the protein structure. Were any specific parameters used to measure this stabilization in the predictive analysis? Please add the SASA values or other relevant measures in brackets next to the sentences where stabilization is explained.

6. In Section 3.9, the authors explained that genes with subsets of mutations from arginine to lysine were analyzed using Gene Ontology (GO) pathways, and these mutations were found to be enriched in the "cell periphery" GO component category. However, the methodology for this analysis is unclear. The authors should provide more details on how the "cell periphery" pathway was identified. Was the most significant pathway selected?

7. In the conclusion, on line 623, the authors cited a paper suggesting that Arg/Lys rebalancing could be tested with wild-type and arginine-depleted membrane proteins in a cell-based assay. Please provide more experimental results supporting the role of the Arg/Lys rebalancing mechanism in maintaining proteostasis in cancer cells, along with relevant citations.

6. PLOS authors have the option to publish the peer review history of their article (what does this mean?). If published, this will include your full peer review and any attached files.

Reviewer #1: No

Reviewer #2: **Yes: **Tamil Iniyan Gunasekaran, Columbia University, United States

---

## [Author Response · Author response to Decision Letter 0]

11 Oct 2024

PONE-D-24-29679

Computational investigation of missense somatic mutations in cancer and potential links to pH-dependence and proteostasis

PLOS ONE

Dear PLOS ONE,

We record our responses to the Reviewer’s comments. We believe that we have addressed all the comments, we hope satisfactorily. We thanks the Reviewer’s for their time and constructive comments.

Jim Warwicker and Shalaw Sallah, Manchester, 11 Oct 2024

Reviewer #1: Review Report:

The paper explores the role of somatic mutations, with a particular focus on arginine depletion and pH-sensing mutations, in cancer cell adaptation to acidic environments and their potential impact on tumor progression. The study employs advanced computational tools, including AlphaFold2 and pKa prediction methods, to analyze the structural and functional implications of these somatic mutations.

The paper highlights potential therapeutic opportunities by targeting pH-sensing mechanisms in cancer cells.

Major Comments:

1. The paper primarily focuses on histidine and arginine mutations, neglecting other potentially relevant mutations that could provide a more comprehensive understanding of cancer cell adaptation. Expanding the analysis to include other amino acid mutations that may play a role in pH-dependent processes would offer a more holistic understanding of the mechanisms involved.

Response: After Fig 3, that presents the Arg and His calculations, results of a new set of calculations are given in S1 Fig. These capture Asp, Glu, and Lys sites of missense mutation in COSMIC and are again sampled at low (1) and higher (≥10) instances. As seen for Arg and Lys sites, there is very little difference in the distribution of calculated �pKas and SASA values between the lower and instance subsets. This analysis is discussed in new text at the end of Section 3.2, with the conclusion that (again) there are no clear systematic differences in the electrostatic environments of the different COSMIC instance sets, and thus no obvious link to alteration in pH-dependent properties. It is noted that smaller subsets may still be relevant in regard of adaptation to pH changes.

2. The introduction and background sections do not provide enough context on the broader landscape of somatic mutations in cancer, limiting the reader's understanding of the study's significance. Enhancing these sections to include a more detailed discussion of the general landscape of somatic mutations in cancer will help readers better appreciate the study's relevance and significance.

Response: Text and references have been added to the Introduction, with a very short history of the study of mutational signatures and their links with mechanisms, including recent coupling to measured genome topological properties. The common underlying endogenous process of methylcytosine deamination, contributing to Arg depletion at the amino acid level is noted, and referenced.

3. The paper places too much emphasis on computational data without adequately discussing the limitations and potential inaccuracies of these methods. A more balanced discussion is needed, critically evaluating the limitations of the computational methods used and discussing potential inaccuracies and their impact on the study's conclusions.

Response: Asking for more discussion of the limitations involved with our computational methodology is reasonable. With regard to the calculations of solvent accessible surface area (SASA) and pKas, these are allowing us to assess possible sites of pH-dependence. We already note in the Introduction that both of the pKa calculation methods (PROPKA3 and pkcalc) have reports of benchmarking against experimental in the literature. To give context beyond this we add text to section 3.5 that discusses 5 systems where biophysical analysis demonstrates a pH-dependence, which in turn may be related to cancer. These 5 mutations are omitted from our Table 1 list of potentially pH-dependent groups because they either do not have 10 instances in COSMIC, or they are already well-characterised. They are though used to test our computations. Three of the 5 are buried with large predicted pKa changes in the AlphaFold protomer models. Both of the other two are involved in functional protein-protein interactions that change environment of the mutation sites, and are likely to mediate the pH-dependence. This demonstrates both the efficacy of our method for identifying pH-dependent sites, but also its limitations where these sites develop at interfaces. The advent of AlphaFold models for whole proteome modelling is itself revolutionary for the field, and it can be anticipated (as referenced in the new text) that proteome interactome analysis with vastly increased coverage (through experimental structures and modelling) will also be a great step forward, although beyond the scope of the current study.

Minor Comments:

1. The introduction should be expanded to provide more context on the broader landscape of somatic mutations in cancer, which is crucial for understanding the study's significance.

Response: See response to major point 2 from this Reviewer, outlining the additional Introduction text and references added.

2. The results are presented clearly, but the discussion does not adequately address the computational methods' limitations. Additionally, the focus on histidine and arginine mutations is too narrow.

Response: In respect of the computational methods’ limitations, please see the response to major point 3 from this Reviewer.

With regard to the focus on histidine and arginine, see response to major point 1 from this Reviewer. We have added Asp, Glu, Lys analysis, and a supplemental Figure to mirror that for Arg and His (Fig 3). Very similar results are obtained for Asp, Glu, Lys to those for Arg, His, specifically that we see no general electrostatic feature difference between sites at higher instances (≥10) in the COSMIC database, compared with those at a single instance, and therefore no indication of a general signal for pH-adaptation.

The paper provides valuable insights into the role of arginine depletion and pH-sensing mutations in cancer. However, major revisions are required to address the lack of experimental validation, narrow focus, and insufficient contextual background. With these revisions, the study could make a substantial contribution to the field.

.

Reviewer #2: Comments to the Authors:

The study titled "Computational Investigation of Missense Somatic Mutations in Cancer and Potential Links to pH-Dependence and Proteostasis" by Shalaw Sallah and Jim Warwicker provides a thorough analysis of somatic missense mutations, specifically focusing on the arginine-to-histidine substitution, which may contribute to pH-sensing functions. Within the frequently mutated subset, the authors identified mutations in NDST1, the HLA-C chain of the MHC I complex, and the water channel AQP-7 as potential mediators of pH-dependence in cancer cells. Furthermore, they emphasized that rebalancing the arginine-to-lysine ratio is crucial for maintaining proteostasis in peripheral cellular locations and controlling tumor development. However, I believe there are areas where further improvements could be made.

Major revisions:

1. In the methods section, the authors noted their use of COSMIC database version 97. However, the latest version, 100, has been released, including several new missense mutations. Did the authors consider analyzing these recently added mutations in the current study? If not, it would be beneficial to include this analysis.

Response: This is a reasonable observation, and applies to bioinformatics analysis in many areas that rely on developing genomics data resources. For a specific response to the Reviewer’s comments we have taken the data used in Table 1, and analysed how these data change for the current (October 2024) COSMIC set, as compared with that for our analysis throughout the work (February 2023 COSMIC dataset). We find, as expected, that there are increases in number of recorded instances for some, but not all, of the 10 mutations listed in Table 1. For 6 mutations at buried sites and with 9 instances in the earlier COSMIC dataset, just one is now at 10 instances (the threshold for Table 1), the remaining 5 staying at 9. It is concluded that analysis will depend in detail on the underlying dataset, but that there are not substantial differences between analyses on datasets compiled 20 months apart. The new text is at the end of section 3.3.

2. In Table 1, ten mutations to histidine are listed along with mutation specificity and pKa values. However, the parameters used to analyze the stability of the protein structures are not clearly defined. Please provide the exact SASA (Solvent Accessible Surface Area) values for each mutation. Additionally, include the SIFT, PolyPhen, and CADD (Combined Annotation Dependent Depletion) scores for each mutation to enhance clarity.

Response: We have added SASA values to Table 1 (for the wild type residue), along with PolyPhen-2 and SIFT predictions of mutation effect on protein function. Rather than CADD we have also added predictions using the recent AlphaMissense method for predicting mutation effects, which are targeted at amino acid changes, and take into account sequence as well as the AlphaFold coverage in modelling structure. References for these are now included in the Methods section, with information on the thresholds used in each prediction scheme included in the Table 1 footnotes. Further additional text in sections 3.3 and 3.4 integrates discussion of the added data in Table 1.

3. In Section 3.7, the authors described how the lysine-to-arginine balance could be beneficial for maintaining cellular proteostasis in tumor cells. Have the authors analyzed Lysine/Arginine (Lys/Arg) mutations in previously published studies on centenarians or healthy aging? Does this Lys/Arg balance contribute to healthy aging, potentially acting oppositely to the mechanism observed in cancer cells?

Response: This is an interesting suggestion that we have researched. We did not find clear-cut data on the role of Arg depletion, or more generally that of somatic mutations, in ageing. We have though included a note towards the end of the Conclusions section that briefly discusses this potentially interesting link.

Minor Revisions:

4. The authors discussed the significance of the arginine-to-histidine mutation in cancer cells, highlighting its role in gaining pH-sensing function. They also referenced previously published studies explaining the mechanism behind this mutation. However, are there any other significant functions gained by this arginine-to-histidine mutation specifically in cancer cells beyond pH-sensing? If so, please elaborate on these in the introduction.

Response: We have researched the literature in this area and can find no other directly appropriate material. It is noted that we have expanded discussion generally of mutational signatures in the Introduction, in response to Reviewer 1 comments. In addition, just after that expanded discussion, we have added a further reference with respect to a potential role for Arg to Cys mutations in alleviating stress from reactive oxygen species.

5. In Section 3.4.2, the authors discussed the stabilization of the protein structure. Were any specific parameters used to measure this stabilization in the predictive analysis? Please add the SASA values or other relevant measures in brackets next to the sentences where stabilization is explained.

Response: We have added text in this section to clarify that we are discussing the electrostatic stabilisation that is predicted to occur for the D268H mutation of PCDHGB4 (in the absence of calcium binding at the site), and that this stabilisation is assessed from the predicted pKa for D268H. Calculated SASA changes little between wild type D268 (16.2 Å2) and mutated 268H (17.6 Å2).

6. In Section 3.9, the authors explained that genes with subsets of mutations from arginine to lysine were analyzed using Gene Ontology (GO) pathways, and these mutations were found to be enriched in the "cell periphery" GO component category. However, the methodology for this analysis is unclear. The authors should provide more details on how the "cell periphery" pathway was identified. Was the most significant pathway selected?

Response: We have added text to section 4.9 specifying that the GO component classification was used in the Princeton GO Finder tool, to identify the most significant categories. Then cell periphery was returned as the most significant (10 of 14 from Arg and to Lys categories in Fig 9), or close to the most significant (the remaining 4), as now stated in section 4.9.

7. In the conclusion, on line 623, the authors cited a paper suggesting that Arg/Lys rebalancing could be tested with wild-type and arginine-depleted membrane proteins in a cell-based assay. Please provide more experimental results supporting the role of the Arg/Lys rebalancing mechanism in maintaining proteostasis in cancer cells, along with relevant citations.

Response: We have provided some more text on the study the Reviewer mentions, in the Conclusions section. That study looked at proteostasis at the cell surface in response to membrane protein cross-linking with antibodies. We suggest that similar methodology could be employed to look at proteostasis effects when Arg / Lys balance is altered for cell surface proteins. We take the point that we have no clear-cut experimental validation of our hypothesised link between Arg / Lys balance and the behaviour of cell surface proteins in cancer. What we do have is a background experimental literature (referenced in the manuscript) on the different properties of Arg and Lys in respect of protein solubility, alongside our bioinformatics observations for cancer mutations. We hope that this report will stimulate experimental work in the area.

We thank both Reviewers for their helpful comments, and hope that our revisions are a suitable response to their suggestions.

---

## [Decision Letter · Decision Letter 1]

5 Nov 2024

Computational investigation of missense somatic mutations in cancer and potential links to pH-dependence and proteostasis

PONE-D-24-29679R1

Dear Dr. Warwicker,

We’re pleased to inform you that your manuscript has been judged scientifically suitable for publication and will be formally accepted for publication once it meets all outstanding technical requirements.

Kind regards,

Rajesh Kumar Pathak, Ph.D.

Academic Editor

PLOS ONE

Additional Editor Comments (optional):

The manuscript can be accepted for publication.

Reviewers' comments:

Reviewer's Responses to Questions

**Comments to the Author**

1. If the authors have adequately addressed your comments raised in a previous round of review and you feel that this manuscript is now acceptable for publication, you may indicate that here to bypass the “Comments to the Author” section, enter your conflict of interest statement in the “Confidential to Editor” section, and submit your "Accept" recommendation.

Reviewer #1: All comments have been addressed

Reviewer #2: All comments have been addressed

2. Is the manuscript technically sound, and do the data support the conclusions?

Reviewer #1: Yes

Reviewer #2: Yes

3. Has the statistical analysis been performed appropriately and rigorously? 

Reviewer #1: Yes

Reviewer #2: Yes

4. Have the authors made all data underlying the findings in their manuscript fully available?

Reviewer #1: Yes

Reviewer #2: Yes

5. Is the manuscript presented in an intelligible fashion and written in standard English?

Reviewer #1: Yes

Reviewer #2: Yes

6. Review Comments to the Author

Reviewer #1: The authors have addressed the major concerns in a comprehensive manner, expanding their analysis to include additional amino acid mutations such as Asp, Glu, and Lys. This additional analysis is well integrated into the manuscript, offering a more complete understanding of pH-dependent processes in cancer. They have also provided new background information on mutational signatures, including references to recent studies, enhancing the contextual understanding of somatic mutations in cancer. Furthermore, the limitations of the computational methods used in the study are now better discussed, and the addition of experimental validations or links to known biophysical systems offers a more balanced perspective. The revised manuscript now provides a clearer and more robust contribution to understanding the role of pH-dependence and somatic mutations in cancer, and I recommend the manuscript for acceptance.

Reviewer #2: Dear Authors,

I hope this message finds you well.

I have reviewed the revised version of your manuscript titled “Computational investigation of missense somatic mutations in cancer and potential links to pH-dependence and proteostasis”. I am pleased to confirm that you have successfully addressed all the revisions and comments I provided in my last review. The changes have significantly improved the quality and clarity of the manuscript.

I have no further comments or suggestions. Thank you for your efforts in thoroughly revising the manuscript.

7. PLOS authors have the option to publish the peer review history of their article (what does this mean?). If published, this will include your full peer review and any attached files.

Reviewer #1: No

Reviewer #2: No

---

## [Editor Report · Acceptance letter]

8 Nov 2024

PONE-D-24-29679R1 

PLOS ONE

Dear Dr. Warwicker, 

I'm pleased to inform you that your manuscript has been deemed suitable for publication in PLOS ONE. Congratulations! Your manuscript is now being handed over to our production team.

Kind regards, 

on behalf of

Dr. Rajesh Kumar Pathak 

Academic Editor

PLOS ONE